# The Functionality Verification through Pilot Human Subject Testing of MyFlex-δ: An ESR Foot Prosthesis with Spherical Ankle Joint

Johnnidel Tabucol [1,2,3,*], Vera Geertruida Maria Kooiman [4,5,*], Marco Leopaldi [2], Tommaso Maria Brugo [1,2], Ruud Adrianus Leijendekkers [4,5,6], Gregorio Tagliabue [3], Vishal Raveendranathan [3], Eleonora Sotgiu [1], Pietro Benincasa [1], Magnus Oddsson [7], Nico Verdonschot [6,8], Raffaella Carloni [3] and Andrea Zucchelli [1,2,*]

1   Department of Industrial Engineering, University of Bologna, 40131 Bologna, Italy; tommasomaria.brugo@unibo.it (T.M.B.); eleonora.sotgiu8@gmail.com (E.S.); benincasa.pb@gmail.com (P.B.)
2   Interdepartmental Centre for Industrial Research in Advanced Mechanical Engineering Applications and Materials Technology, University of Bologna, 40131 Bologna, Italy; marco.leopaldi@unibo.it
3   Bernoulli Institute for Mathematics, Computer Science and Artificial Intelligence, Faculty of Science and Engineering, University of Groningen, 9747 AG Groningen, The Netherlands; g.tagliabue@rug.nl (G.T.); v.raveendranathan@rug.nl (V.R.); r.carloni@rug.nl (R.C.)
4   Orthopedic Research Laboratory, Radboud University Medical Center, 6525 GA Nijmegen, The Netherlands; ruud.leijendekkers@radboudumc.nl
5   Department of Rehabilitation, Donders Institute for Brain, Cognition and Behavior, Radboud University Medical Center, 6525 GA Nijmegen, The Netherlands
6   IQ Healthcare, Radboud Institute for Health Sciences, Radboud University Medical Center, 6525 GA Nijmegen, The Netherlands; nico.verdonschot@radboudumc.nl
7   Research and Development, Össur, 110 Reykjavík, Iceland; magnuso@ossur.com
8   Department of Biomechanical Engineering, University of Twente, 7522 NB Enschede, The Netherlands
*   Correspondence: johnnidel.tabucol2@unibo.it (J.T.); vera.kooiman@radboudumc.nl (V.G.M.K.); a.zucchelli@unibo.it (A.Z.); Tel.: +39-320-8177937 (J.T.); +39-348-4559575 (A.Z.)

**Abstract:** Most biomechanical research has focused on level-ground walking giving less attention to other conditions. As a result, most lower limb prosthesis studies have focused on sagittal plane movements. In this paper, an ESR foot is presented, of which five different stiffnesses were optimized for as many weight categories of users. It is characterized by a spherical ankle joint, with which, combined with the elastic elements, the authors wanted to create a prosthesis that gives the desired stiffness in the sagittal plane but at the same time, gives flexibility in the other planes to allow the adaptation of the foot prosthesis to the ground conditions. The ESR foot was preliminarily tested by participants with transfemoral amputation. After a brief familiarization with the device, each participant was asked to wear markers and to walk on a sensorized treadmill to measure their kinematics and kinetics. Then, each participant was asked to leave feedback via an evaluation questionnaire. The measurements and feedback allowed us to evaluate the performance of the prosthesis quantitatively and qualitatively. Although there were no significant improvements on the symmetry of the gait, due also to very limited familiarization time, the participants perceived an improvement brought by the spherical ankle joint.

**Keywords:** gait analyses; prosthetic foot; transtibial amputation

## 1. Introduction

In this paper, the authors present Myflex-δ, an energy-storing-and-releasing (ESR) foot prosthesis with a spherical joint as ankle joint. The goal of the spherical joint at the ankle is to ensure the adaptation of the foot to the irregular conditions of the ground. Since the spherical joint is applied to an ESR foot, this adaptation is done passively, without the use of external energy source, i.e., battery.

*1.1. Categories of Ankle–Foot Prostheses*

From the point of view of the development of foot prostheses, several studies have brought excellent results from a technological point of view with the integration of actuators. Actuators have enabled prosthetic devices to change their characteristics and configurations to adapt to the conditions of locomotion (without injecting propulsive power), as in the case of foot prostheses defined as semiactive [1–7]; actuators with greater power than those integrated in semiactive prostheses have been integrated into prostheses defined as bionic so as to inject propulsive power to perform different activities such as walking, climbing or descending stairs or ramps [8–43]. In essence, due to their characteristics, semiactive and bionic prostheses are suitable prostheses for amputees with a K3 ambulatory level (active individuals not limited to low-cadence walking as are amputees with a K2 ambulatory level) and K4 (sports individuals). However, they are expensive prostheses. Cost considerations are very important, especially since 80% of potential users reside in developing countries, where 95% of them even struggle to afford the simplest and cheapest prostheses [44]. The most economical prostheses compared to semiactive and bionic and suitable for amputees with a K3 and K4 ambulatory level are ESR feet. With the composite elastic elements they are composed of, ESR feet accumulate elastic energy during the mid stance (subphase of the stance phase that goes from toe strike to heel-off); this energy is then released during the push-off for propulsion [45,46]. Conventional feet are also cheaper than ESR feet, but they are foot prostheses mostly suitable for amputees with a K2 ambulation level, therefore relatively less active users.

*1.2. Reasons to Design Multiaxial Ankle–Foot Prostheses*

According to the literature, patent and market research carried out by the authors, most of the studies on foot prostheses have focused on movement in the sagittal plane. Only some of these studies were developed considering the rotations of the foot in the other two planes, i.e., frontal plane and transverse plane [19,30,31,38,47]. However, they are still under development or in a conceptual phase of the design. The importance of developing a foot prosthesis that also considers the abduction–adduction (rotation in the transverse plane) and the inversion–eversion (rotation in the frontal plane) is justified by the fact that even the healthy foot, during level-ground walking, not only rotates in the sagittal plane, but eversion–inversion and adduction–abduction also occur [48,49]. Rotations in the frontal and transverse planes are even more accentuated in other types of movements such as the turning step or side-stepping, where the external foot with respect to the directions of these two movements passes from being in eversion to being in inversion [47]. These two movements are not to be underestimated, as the turning step movement alone represents about 25% of the daily steps [50]. In addition, it should be considered that people can find themselves walking on laterally inclined surfaces during their daily life. In fact, walking on laterally inclined surfaces is something that people face in everyday, especially because in urban streets, sidewalks can be laterally inclined up to 6° for drainage reasons [51]. Moreover, in general, the surfaces on which individuals walk are not always perfectly flat and regular, therefore, adaptations are always necessary.

The rotations of the foot prosthesis to adapt to the ground conditions depend on the loads acting on it (module and direction) and on the stiffness/flexibility of the foot in the three directions of rotation. A foot prosthesis designed to move only in the sagittal plane has stiffness tending to infinity for rotations in abduction–adduction and inversion–eversion. With infinite rotational stiffness in the transverse plane and frontal plane, any loads on the foot in those two planes generate high torque at the ankle. This high torque is then transferred to the stump of the amputee generating a perception of discomfort or even pain. Designing a prosthesis that also allows rotations at the ankle in the frontal plane and transverse plane does not mean designing a prosthesis with free abduction–adduction and inversion-eversion, but with a stiffness not tending to infinity in the above-mentioned planes.

### 1.3. The Human Ankle Complex

The choice to develop a foot prosthesis with a spherical joint as an ankle joint derives from the study of the human ankle complex, its structure and the movements it allows. The ankle joint complex consists mainly of three joints: the tibiotalar or talocrural joint (formed by the tibia and the talus bones), the talocalcaneal or subtalar joint (formed by the talus and the calcaneus), and the transverse-tarsal or talocalcaneonavicular joint (articulation formed by three bones: talus, calcaneus and navicular). Some studies have suggested that the tibiotalar joint has a geometry indicating that it does not only work as a hinge joint [52,53], even if other studies have suggested that it is a uniaxial joint with the axis oblique to all three planes of motion of the foot-ankle system [53,54]. Most of the dorsiflexion–plantarflexion of the ankle occurs in this joint [55]. The subtalar joint also has an oblique axis and thus generates a triplanar motion [55]. A study suggests that most of the ankle inversion–eversion takes place in the subtalar joint [56]. The transverse-tarsal joint is considered to be part of the same functional unit as the subtalar joint, considering that these two joints share a rotational axis [53,56] and also contributes to the inversion–eversion of the ankle [55].

### 1.4. State of the Art of Multiaxial Foot Prostheses

The main plane of locomotion is the sagittal plane. For this reason, most studies of biomechanics in general and lower limb prostheses in particular have focused almost exclusively on the sagittal plane. However, the idea of adding the possibility of rotating a foot prosthesis in the sagittal plane is not entirely new. Indeed, several designs of foot prostheses that ensure multiaxiality have already been patented or commercialized [57–66]. Furthermore, academically, prosthetic devices with additional rotations in the frontal and transverse planes have been developed [19,30,31,38,47]. In previous studies and/or devices, different approaches were followed to address the same necessity of creating flexibility in all directions of rotation at the ankle–foot prosthesis. In Sections 1.4.1 and 1.4.2, these different design approaches are briefly described.

### 1.4.1. Commercial Devices and Patents

In several designs, elastomeric materials were used together with composite components to create the desired flexibility of the foot prosthesis in sagittal, frontal and transverse planes [57–61,67–69]. Split geometries are used in current commercial ESR feet: the elastic parts of the foot prosthesis are cut partially in the longitudinal direction to allow slight eversion and inversion of the foot in case of uneven terrain or laterally sloped grounds. Special modules that create internal/external rotation of the foot prosthesis, besides the function of absorbing the impact of the foot with the ground, are also used; these modules are added externally to ESR feet, such as Elite$^2$ VT and Echelon VT by Blatchford (blatchford.co.uk, accessed on 2 January 2022), the Pro-Flex LP Torsion and the Pro-Flex XC Torsion feet by Össur (www.ossur.com, accessed on 2 January 2022) and the Taleo Harmony and Triton VS feet by Ottobock (www.ottobock.com, accessed on 2 January 2022). These modules are typically placed between the pylon connection and the foot or ankle (if the prosthesis has an ankle joint). A different approach is used with the Triton Side Flex foot by Ottobock, which guarantees a rotation in the frontal plane by means of a torsion bar inside a system mounted externally to the elastic group of the ESR foot. The use of the spherical joint in ankle–foot prostheses to have a multiaxial ankle joint has been seen almost only among conventional feet. The spherical joint creates the possibility for the foot to move in all directions and elastomeric bumpers [62–66] or springs [67–69] are used to create torsional stiffness and/or damping.

### 1.4.2. Academic Research

Academically, some previous works about foot prostheses have considered a multiaxial ankle. Bellman et al. realized SPARKY 3, a prosthetic foot with actuated motions in the sagittal and frontal planes [19]. Seeing in their studies that the rotation of the ankle in inversion/eversion significantly changed during turning maneuvers when compared to straight

walking, Ficanha and Rastgaar developed a prosthetic foot with two degrees of freedom, dorsiflexion/plantarflexion and eversion/inversion, both actuated [47]. Madusanka et al. designed and tested a foot prosthesis with three degrees of freedom at the ankle joint, where the dorsiflexion/plantarflexion and adduction/abduction are active, leaving the inversion/eversion passive [30]. A spherical joint was used by Masum et al. in a conceptual design of an ankle system in which the rotation in the sagittal plane could be actuated and the rotation in the frontal plane was passively actuated by means of springs and dampers [31]. Agboola-Dobson proposed a novel powered ankle–foot prosthesis which provides two degrees of freedom motions at the ankle joint: thanks to a custom U-joint ankle joint, extrarotations in the frontal plane are permitted, while the motion in the sagittal plane is actuated through a series elastic actuator that emulates the biomechanics of the calf muscle and Achilles tendons [38]. All these prostheses are still in the development phase or still in the conceptual design phase, therefore not yet in use.

### 1.4.3. MyFlex-$\delta$ vs. Previous Technologies

Based on Sections 1.4.1 and 1.4.2, the multiaxiality of the ankle system is guaranteed by the deformation of the elastic elements of ESR feet cut longitudinally (rotations in the frontal and transverse planes occur at the foot and not around the ankle), by external modules in some commercial prostheses (rotations in the frontal and transverse planes occur above the ankle), by a torsion bar inserted into a system at the ankle level of an ESR prosthesis, by a spherical ankle joint in conventional feet where stiffnesses in the various directions of rotation are created by means of springs or elastomeric elements or by other types of solutions on bionic prostheses, which are however still under development. Therefore, it can be stated that no prosthesis belonging to the category of ESR feet is characterized by a spherical ankle joint. In this project, the authors applied the spherical ankle joint to a prosthesis that belongs to the category of ESR feet with the aim of providing a prosthesis that is: (i) suitable for users with a K3 or K4 level of ambulation, (ii) adaptable to passive ground conditions and (iii) relatively low cost. Bionic feet and semiactive feet, also suitable for users with K3 and K4 ambulation levels, may be too expensive for most potential users, as mentioned previously. The novelty, from the biomechanical-prosthetic point of view is therefore the introduction of the spherical ankle joint in an ESR foot, not to mention the purely mechanical aspect as the spherical joint was made of carbon fiber composite.

### *1.5. Functionality Evaluation of Foot Prostheses*

To evaluate the functionality of MyFlex-$\delta$, whose features are described in Section 2.1.1, two types of tests were carried out. There are different ways of assessing the biomechanics of human walking in general and of walking with ankle–foot prostheses, and Hansen collected and presented them in [70]. In [70], Hansen cited two methods for determining the functional performance of a foot prosthesis: one method consisted of testing the device without interaction with humans (mechanical properties testing), while the other consisted of having the prosthesis worn by amputees (human subjects testing) and having them perform one or more activities.

### 1.5.1. Mechanical Properties Testing

In some mechanical tests, the foot prostheses are loaded at different angles of inclination that correspond to the angle of the shank with respect to the ground during the walk [71–73]. In other works, the foot prosthesis is tested statically using the procedure given in ISO 10328 standard to determine the stiffness of the foot and to determine its strength to more critical loads [74], or dynamically following the procedure given in ISO 22675 [4]. The American Orthotic and Prosthetic Association (AOPA—www.aopanet.org, accessed on 2 January 2022) offers a guideline to perform both static and dynamic tests on foot prostheses, both in the sagittal plane and in the other two planes, transverse and frontal.

1.5.2. Human Subjects Testing

In tests with human subjects, the performance of foot prostheses can be evaluated qualitatively, through questionnaires, or quantitatively, through direct or indirect measurements of kinematics and kinetics. The user can give feedback regarding the benefits or negative sides of the prosthesis and/or answer questions regarding the perceived sensations or even the aesthetics of the device [75–83]. In [70], Hansen identified and cited the gait analysis on level ground [71,72,76,78–80,84–106] and stair [95] and ramp ambulations [78,85,93] as the most common forms of motion assessment.

*1.6. Summary*

MyFlex-$\delta$ was developed and optimized following a design methodology presented by Tabucol et al. in [74]. Five prototypes of MyFlex-$\delta$ were produced optimizing five stiffnesses for as many weight categories of users: 60 kg, 70 kg, 80 kg, 90 kg and 100 kg. The final stiffnesses of the five samples, to confirm also the results of the finite element analyses during the optimization, were subsequently determined with static tests which were equivalent to the static tests proposed both in the ISO 10328 standard and AOPA guidelines. Static and fatigue tests at critical loads were subsequently carried out to certify these prostheses from a structural point of view, following the ISO 10328 test procedure. Once the certification was obtained for functional tests with users, human subjects testing with three participants with transfemoral amputation were performed, with all participants having different weights. Results of the MyFlex-$\delta$ were compared to the participants daily prosthesis. Being compared with a device that is used every day by the participants, from these tests, the authors did not expect a great improvement from the point of view of gait symmetry with MyFlex-$\delta$, but at least a maximum difference of 10% in the evaluation parameters was the goal. A possible improvement was expected by the authors on the perceptions of the participants regarding the walks on uneven terrain and changes of direction, thanks to the spherical ankle joint. Therefore, the aim of the human subject testing was to evaluate gait symmetry and ankle movements during level walking with the MyFlex-$\delta$ in persons with a transfemoral amputation compared to participant's current prosthetic foot. In addition, participants' experience with the MyFlex-$\delta$ on uneven and even terrain was subjectively evaluated. In the continuation of the reading of this paper, the reader will find in sequence the brief presentation of the mechanical structure of MyFlex-$\delta$ (Section 2.1.1), its working principle (Section 2.1.2), a very short description of the design methodology followed (Section 2.2) and then a more accurate description of the human subjects testing performed with the participants (Section 1.5.2). Finally, the results of human subjects testing are reported (Section 3) and discussed (Section 4).

**2. Materials and Methods**

In this section, the mechanical design and working principle of Myflex-$\delta$ (Section 2.1), the mechanical tests used to validate the results obtained from the FEM simulations with which the five stiffnesses were optimized (Section 1.5.1) and the human subjects testing (Section 1.5.2) carried out with the three participants are presented in sequence.

*2.1. The Mechanical Design and Working Principle of MyFlex-$\delta$*

The following sections describe the mechanical design (Section 2.1.1) and working principle in the sagittal (Section 2.1.2) of MyFlex-$\delta$. In addition, a brief description of how the elastic elements contribute to the stiffness of MyFlex-$\delta$ in all directions is given (Section 2.1.3).

2.1.1. Mechanical Design of MyFlex-$\delta$

MyFlex-$\delta$ can be divided into subgroups (Figure 1a). The foot subgroup consists of the three elastic elements (upper blade, middle blade and lower blade) joined together at the metatarsal level by two bolts. The spring holder is also part of the foot subgroup. The tendon subgroup consists mainly of the link part. In detail, the link consists of two hinge

parts, one upper and one lower. The third subgroup is the ankle subgroup, consisting mainly of the ankle frame, spherical ankle joint and tube connector. The spherical ankle joint is composed of the spherical component and its seats, which are made of two parts (one upper seat and one lower seat). The seats are fixed to the ankle frame by means of screws. The three components of the spherical ankle joint are made of carbon fiber composite material. The foot subgroup and the tendon subgroup are connected via the spring holder and the lower link. The connection between the spring holder and the lower link is made through a hinge joint, while the connection between the spring holder and the middle blade is made through bolts. The upper link is connected to the tube connector in the same way the lower link is connected to the spring holder, i.e., by means of a hinge joint. Finally, the ankle subgroup is connected to the foot subgroup by two screws.

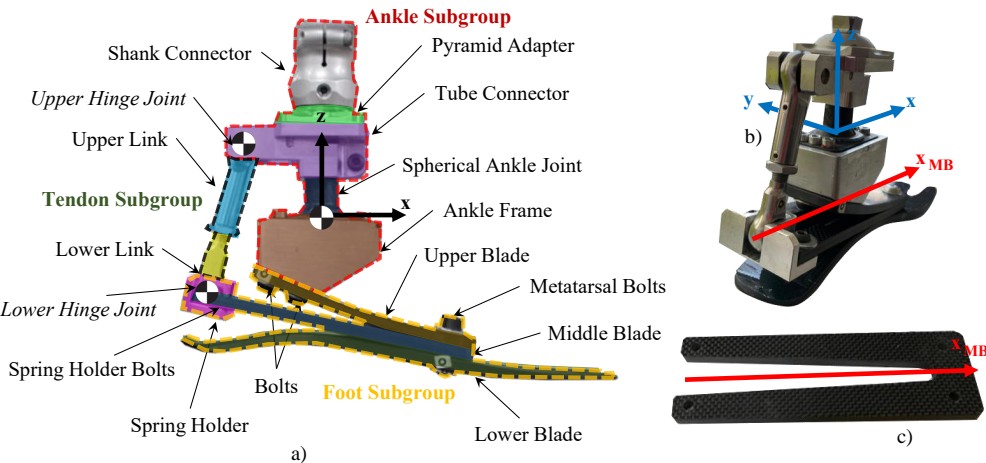

**Figure 1.** (**a**) MyFlex-$\delta$ parts and subgroups; (**b**) picture of MyFlex-$\delta$; (**c**) shape of the middle blade.

The spherical ankle joint combined with the elastic elements, and in particular with the middle blade, allows the foot not only to rotate in the sagittal plane but also in the other two (transverse and frontal). Its fork shape (Figure 1c) allows the middle blade to also have torsional deformations in the frontal and transverse planes, allowing the inversion/eversion and abduction/adduction rotations of the foot.

2.1.2. Working Principle of MyFlex-$\delta$ in the Sagittal Plane

Concerning the configuration of the foot prosthesis, during the heel strike, the heel portion of the lower blade and the middle blade absorb the impact with the ground. The contribution of the upper blade during a heel strike is minor compared to the contribution of the middle blade and the lower blade. The foot rotates in plantarflexion to direct with respect to the ankle the force exchanged between the foot and the ground (ground reaction force). The plantarflexion movement continues until toe strike occurs (when the toe touches the ground). From this stage onward, thanks to the sum of forces given by the inertia, by the push that the prosthesis user gives with the opposite leg and by the elastic energy previously accumulated by the lower blade and the middle blade, the foot prosthesis returns first in the resting configuration, and then deform in the opposite direction, generating a dorsiflexion of the foot. From toe strike, the foot is loaded at both heel and toe. From the toe strike to the heel-off (when the heel comes off the ground), a load transfer from the heel to the toe occurs. When the force given by the weight of the body and the inertia, is in front of the vertical line that passes through the ankle, the foot begins to flex. The upper blade and middle blade deform by bending, and while bending, they accumulate elastic energy. The elastic energy that is accumulated during this phase, is then released during the push-off phase.

### 2.1.3. Elements Involved in the Torsional Stiffnesses of the Ankle Joint

The middle blade and the lower blade (rear portion) are the elastic elements involved during plantarflexion, and therefore, the stiffness in this direction of rotation is created by these two elements. The stiffness in dorsiflexion is mainly created by the middle blade and the upper blade with a slight involvement of the lower blade. As the prosthesis is structured, the stiffnesses for rotations in the frontal plane and transverse plane are mainly created by the deformation of the middle blade, the shape of which allows twisting of the same element around its local x-axis ($x_{MB}$, as shown in Figure 1). Such twists, depending on how the prosthesis is loaded, then allow adduction–abduction and inversion–eversion rotations around the ankle spherical joint.

### 2.2. The Design Methodology of MyFlex-δ

The design methodology used to optimize the five stiffnesses of as many Myflex-δ prototypes is described in detail in [74]. The methodology of design and functional verification consisted of three fundamental phases: (i) the design phase through FEM analysis in which the FE model of the prosthesis was simulated with loads and constraints specified by ISO 10328, (ii) the validation phase through static tests on the prototype of physical prostheses replicating the loads of the design phase, and (iii) the functional verification phase in which, through FEM simulations, the dynamic functioning of the same prosthesis was simulated. During the design phase, the elastic elements were optimized in terms of both stiffness and strength, while the other components were optimized only in terms of strength.

To optimize the stiffness of the five foot prostheses, the rotations of the healthy human ankle in the sagittal plane and the loads to which the foot is subjected in level-ground walking were considered. In particular, the elastic elements of MyFlex-δ were dimensioned in such a way as to provide a plantarflexion between −5° and −8° when the heel was subjected to a load corresponding to between 95% and 130% of the body weight of the user. As for the dorsiflexion, the rotation range taken as an objective was between 14° and 18°, for rotations to be obtained when the foot was subjected to a load corresponding to between 95% and 108% of the participant's body weight. These values were chosen by the authors taking into account the biomechanical values reported in the literature concerning the walking data of healthy people [49,107–113]. In particular, these loads correspond to the two peaks of the M-shape ground reaction forces. Therefore, the above-mentioned rotations must be guaranteed when the prosthesis is subjected to the loads reported in Table 1.

**Table 1.** Range of heel and toe loads that must ensure a plantarflexion between −5° and −8° and a dorsiflexion between 14° and 18°, respectively, for different weight categories.

| | Heel Load (Plantarflexion) | | Toe Load (Dorsiflexion) | |
|---|---|---|---|---|
| Weight Categories | 95% | 130% | 95% | 108% |
| 60 kg | 559 N | 765 N | 559 N | 636 N |
| 70 kg | 652 N | 893 N | 652 N | 742 N |
| 80 kg | 746 N | 1020 N | 746 N | 848 N |
| 90 kg | 839 N | 1148 N | 839 N | 954 N |
| 100 kg | 932 N | 1275 N | 932 N | 1059 N |

### 2.2.1. Design Phase

The design phase is the first phase of the design methodology and it consists of two subphases: (a) the geometry optimization phase through 2D FEAs and (b) the material properties optimization phase through 3D FEAs. The 2D FEM model of the foot prostheses were simulated statically by considering the equivalent loads and constraints of the ISO 10328 static dorsiflexion and plantarflexion tests. This procedure was followed in order to perform preliminary optimization of the geometry (geometry optimization subphase).

Subsequently, once the geometry had been preoptimized, a 3D FEM model of the foot prosthesis was simulated statically, according again to equivalent loads and constraints of the ISO 10328 static tests. With the FEM 3D model, the properties of the materials composing the elastic elements were optimized to achieve the desired final stiffness (material properties optimization subphase).

### 2.2.2. Validation Phase through ISO 10328-Equivalent Static Tests

Once the five prototypes with five different stiffnesses were optimized, they were subsequently manufactured and tested both to confirm the stiffness obtained in the FEAs, and to verify the structural integrity with critical loads. The tests carried out on the five examples of MyFlex-$\delta$ were again static and equivalent to the standard static tests regulated by the ISO 10328 standard. The two static tests are shown in Figure 2. In static tests, the foot, fixed at the top of the shank connector (to the load cell), is compressed by a platform to which a force is set. In the plantarflexion test, the foot is relatively inclined backwards by 15° compared to the platform to simulate the early stance that starts from a heel strike and goes up to a toe strike. In the dorsiflexion test, the foot is relatively inclined forward by 20° compared to the platform to simulate the inverted midstance, i.e., the test starts with the toe touching the platform and with the foot inclined but discharged (toe-off, end of midstance) and ends with the heel and toe loaded (heel-off, beginning of midstance). In the plantarflexion and dorsiflexion tests, the relative inclinations between the shank and the platform remain the same. The vertical displacement of the platform is measured during the test, as well as the load (reaction force), measured by the load cell, to which the foot is subjected while being compressed by the platform. The results of the static tests can be provided as reaction force–foot rotation (see Section 3.1.1), where the foot rotation is calculated as the variation of the angle formed by the HM markers and the shank axis. The same results can also be provided as reaction force–platform displacement (see Section 3.1.2). The graphs reaction force–foot rotation are useful and easier to interpret from a biomechanical point of view, while to calculate the elastic energy stored by the elastic elements the graphs reaction force–platform displacement are used (Section 2.2.4).

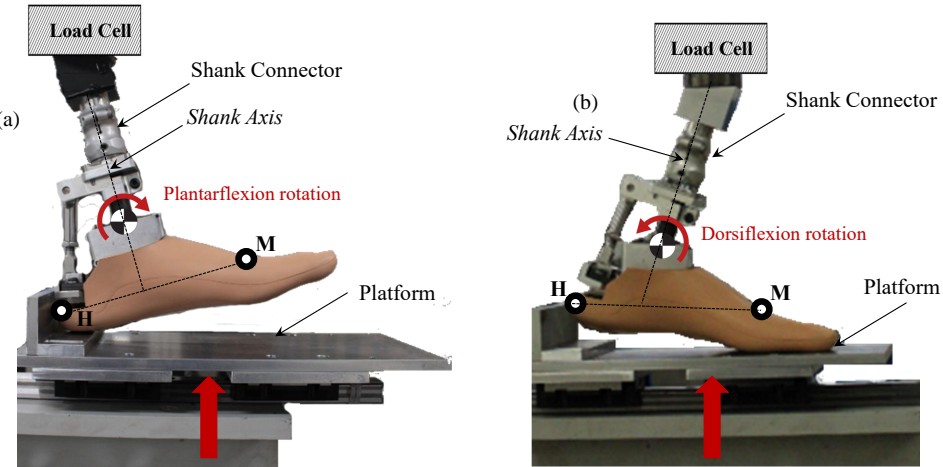

**Figure 2.** The ISO 10328 equivalent static test: (**a**) plantarflexion test and (**b**) dorsiflexion test (right).

### 2.2.3. Functionality Verification Phase

Finally, transient simulations were performed following the dynamic tests proposed in ISO 10328 and ISO 22675. All FEM simulations were performed on Ansys Workbench 2019R3.

### 2.2.4. Elastic Energy Calculation

The AOPA guideline suggests a way to calculate the elastic energy stored by the prosthesis using the reaction force–platform displacement curve. The energy that is accu-

mulated during the midstance (from toe strike to heel-off) being of particular interest, for this work, only the results of the static tests for the dorsiflexion are reported. To calculate the stored elastic energy, following the procedure suggested by the AOPA guideline, the area under the reaction force–platform displacement curve was calculated using the trapezoidal method. The guideline suggests to calculate the area up to 1230 N. However, for the aim of this work, for all curves, the stored energy was calculated only up to 108% of the weight category of the prosthesis, considering the corresponding maximum ground reaction force before heel-off. This means that for the 60 kg optimized prosthesis, the energy was calculated up to 636 N; for the 70 kg prosthesis, the energy was calculated up to 741 N, and so on.

### 2.3. The Human Subjects Testing of MyFlex-δ

For the human subjects testing, three participants were included. All participants had a transfemoral amputation on the left leg due to trauma, and their prostheses were attached to their femur through osseointegration. The characteristics of the three participants are displayed in Table 2. Participants were highly active in their daily life and had no problems with their current prosthesis (e.g., alignment, etc.).

**Table 2.** Characteristics of the participants included in the MyFlex-δ functional tests. [a] = Ottobock, [b] = Össur.

|  | Participant 1 | Participant 2 | Participant 3 |
|---|---|---|---|
| Age (years) | 63 | 46 | 61 |
| Gender | M | M | M |
| Years of amputation (years) | 13 | 14 | 10 |
| Height (cm) | 171 | 184 | 181 |
| Weight with prosthesis (kg) | 103.3 | 80.6 | 73.3 |
| Weight of own prosthesis (kg) | 3.7 | 3.1 | 4 |
| Leg length (cm) | 78 | 84 | 86 |
| Prosthetic knee | Genium [a] | C-Leg 4 [a] | Rheo Knee [b] |
| Prosthetic foot | Trias [a] | Trias [a] | Proprio Foot [b] |
| MyFlex-δ tested | 90, 100 kg | 70–90 kg | 60–80 kg |
| K-Level | K4 | K4 | K4 |

All participants provided a written informed consent before participating in the study. The study procedures were approved by the ethical committee CMO Arnhem-Nijmegen (2019–5920) and complied with the guidelines defined in the Declaration of Helsinki. The approval date of the ethical committee was 16 January 2020.

#### 2.3.1. Procedure

The functional evaluation of MyFlex-δ for each of the participant was carried out in one day. Firstly, measurements were done with the participant using their own prosthesis, followed by the introduction and familiarization to the MyFlex-δ, and the measurements using the MyFlex-δ. The functional measurement as well as the participant experience were used to evaluate the MyFlex-δ in comparison to their own prosthesis. Both pre-(using their own prosthesis) and post-measurements (using MyFlex-δ) entailed the same measurement protocol.

During the familiarization period, a prosthetic technician replaced the prosthetic foot of the participant with the MyFlex-δ of the proper stiffness/weight category, with the participants wearing their own powered knee prostheses set for the characteristics and behavior of their foot prosthesis and not for Myflex-δ. The alignment reference line was aimed to be positioned 30 mm posterior to the middle of the foot, which is a common alignment recommendation for commercially available feet (Figure 3). After proper alignment, the participants were able to familiarize themselves with the MyFlex-δ under supervision of a physiotherapist. Various activities were performed, e.g., sit-to-stand, level-

ground walking with (Figure 4a) and without support(Figure 4c), level-ground side walking (Figure 4b), walking on a treadmill, up- and downslope walking on a treadmill (Figure 4d), and walking on uneven terrain (Figure 4e). Introduction and familiarization with the MyFlex-$\delta$ lasted around 4 h. Participants were able to try at least two different MyFlex-$\delta$ stiffnesses and choose the most suitable one, according to their sensations and the physiotherapist's recommendations. The first participant, weighing 100 kg, had two prototypes of MyFlex-$\delta$ tested, those theoretically optimized for 90 kg and 100 kg, preferring the latter. The second participant, whose weight was 80 kg, tested the MyFlex-$\delta$s optimized for 70 kg, 80 kg and 90 kg, choosing the one for 90 kg. Finally, the third chose the prosthesis for 70 kg after testing the MyFlex-$\delta$s for 60 kg, 70 kg and 80 kg.

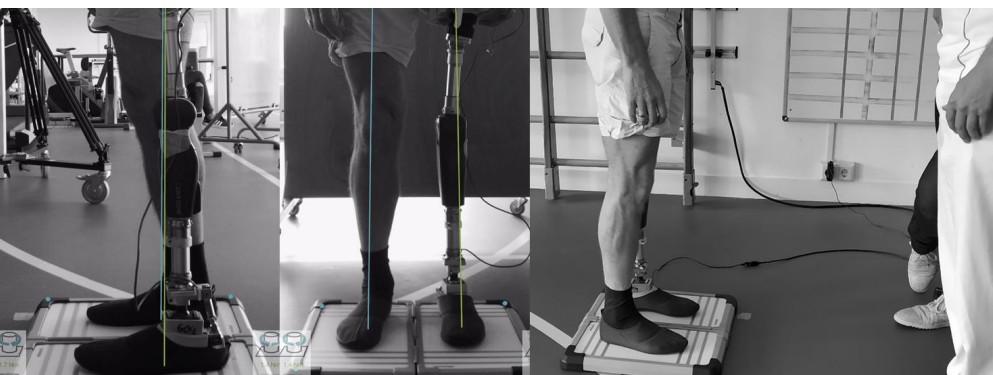

**Figure 3.** Alignment of the foot prosthesis.

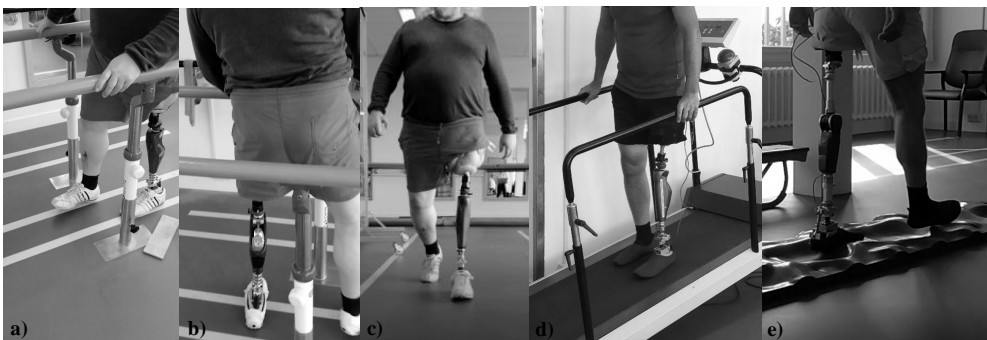

**Figure 4.** Familiarization activities: (**a**) supported level-ground walking; (**b**) supported level-ground side walking; (**c**) level-ground walking; (**d**) supported downslope walking on a treadmill; (**e**) uneven terrain walking.

The measurement protocol consisted of gait analyses, where participants walked for 5 min on a treadmill (with force plates to measure the ground reaction force—Motekforce Link, Amsterdam/Culemborg, The Netherlands), fixed at a self-selected comfortable walking speed. To capture the participant's movements, reflective markers were placed on the participant's skin and prosthesis according to the 'Plug-In Gait Full body' marker model (Plug-in Gait Reference Guide, Vicon Motion Systems Limited, 2021). Three-dimensional movements were captured using the Vicon Nexus system (Vicon Motion Systems, Inc., Lake Forest, CA, USA, 100 Hz). In addition, participants wore a harness for safety (Figure 5). Throughout the whole day, participants were asked about their experience using the MyFlex-$\delta$.

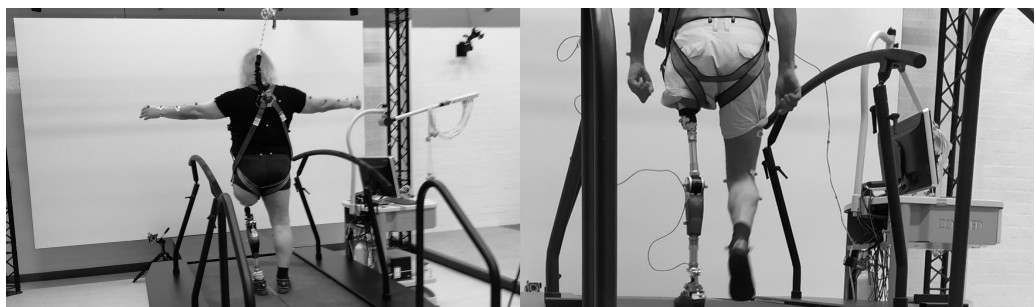

**Figure 5.** Treadmill test: the participant's movements are measured thanks to the reflective markers placed on their skin. The participant wore a harness for safety.

### 2.3.2. Kinematics and Kinetics Data Collection

Vicon Nexus 2.11.0 and Matlab R2019b were used for further data analyses. Data were filtered using a Woltring filter (MSE = 10). Gait events, including heel strike and toe-off, were determined using a ground reaction force threshold of 20 N. The ground reaction force was measured as an analog signal at 2000 Hz. Data were segmented and normalized into data from the heel strike of the right foot to the consecutive heel strike of the right foot. The motion capture system used to measure the kinematics and the force plates used to measure the ground reaction forces were synchronized. As the current study was an initial performance evaluation of the MyFlex-$\delta$ with a small sample size, no statistical analyses were done.

## 3. Results

In this section, the authors present the results from the design and validation phase (Section 3.1) and from the human subjects testing with the three participants (Section 3.2).

### 3.1. Results from Design Phase and Validation Phase

The mechanical properties tests were carried out both to validate the results of the design phase and to characterize in general the five specimens of MyFlex-$\delta$. Section 3.1.1 compares the reaction force–foot rotation curves obtained from the 2D and 3D FEAs and the mechanical properties tests. In Section 2.2.4, the reaction force–platform displacement curves (only dorsiflexion) for all five MyFlex-$\delta$s are shown instead.

### 3.1.1. Comparisons among 2D FEA and 3D FEA Mechanical Properties Testing

In this section, the results from the design phase (Section 2.2.1) and the validation phase (Section 2.2.2) are reported and compared. In particular, the comparison is made through the graphs reaction force–foot rotation, calculated as explained in Section 2.2.2. The results and comparisons are given for both plantarflexion (top) and dorsiflexion test (bottom), as shown in Figure 6.

Since the results of simulations and mechanical properties testing were similar for all five prostheses, for reasons of clarity and comparison, in this section only the results for the prosthesis optimized for a user's weight of 60 kg are shown and compared. The curves of dorsiflexion are very similar to each other, while the same cannot be said for those of plantarflexion. Considering the plantarflexion load condition, the 2D FEA and 3D FEA curves diverge from the test curve obtained from the mechanical properties testing of the prototype. The divergence could be caused by the following reasons: (i) the imperfect mounting of the foot prosthesis on the test setup; (ii) the imperfect CAD and FE modeling of the foot shell; (iii) manufacturing defects that can cause slight variations in the geometry of the physical prototype; (iv) the joints, such as the spherical ankle joint, hinge joints between the link and tube connector, and between the link and spring holder, modeled without friction and thus approximated; (v) the imperfect orientation of the fibers during the production phase of the elastic composite components; (vi) the heel of the foot shell not fully covered by the platform; (vii) the interference between the spring holder part and the

foot shell, as also shown in Figure 7. Since all five prostheses had the same configuration of elastic elements and with difference only in the rolling sequence of the carbon fiber prepreg layers used to produce the same elastic elements, the problem of plantarflexion was found in all five prosthetic devices.

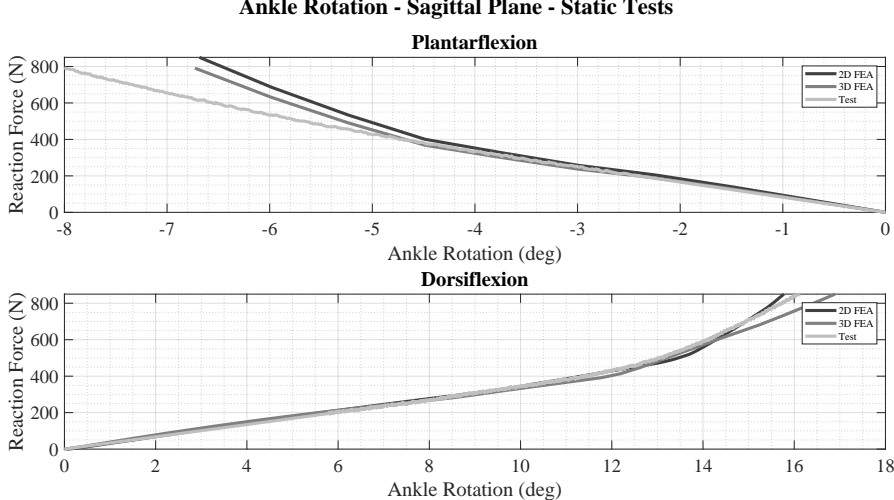

**Figure 6.** Comparisons among the static plantarflexion and dorsiflexion 2D FEAs and 3D FEAs and the mechanical tests on the physical prototype.

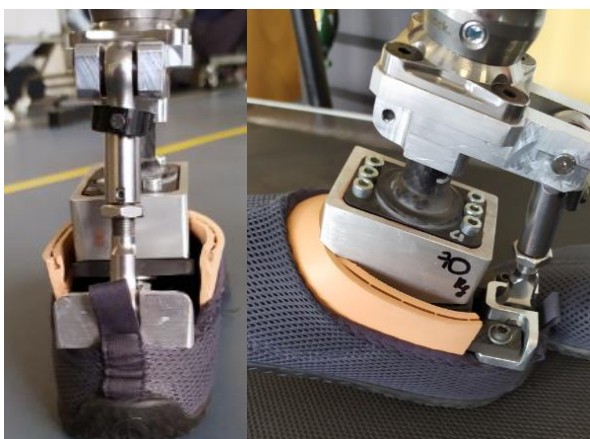

**Figure 7.** The interference between the middle blade and the foot shell (not designed for MyFlex-$\delta$).

### 3.1.2. Calculated Stored Elastic Energy

For the calculation of the elastic energy stored by the elastic elements during the midstance, the energy to be released during the push-off, only the static tests of dorsiflexion were considered. The results of the static tests in dorsiflexion are shown in Figure 8, where at the top they are presented as reaction force–platform displacement, while at the bottom as reaction force–foot rotation. As suggested by the AOPA guideline, for the energy calculation, the reaction force–platform displacement curves were considered.

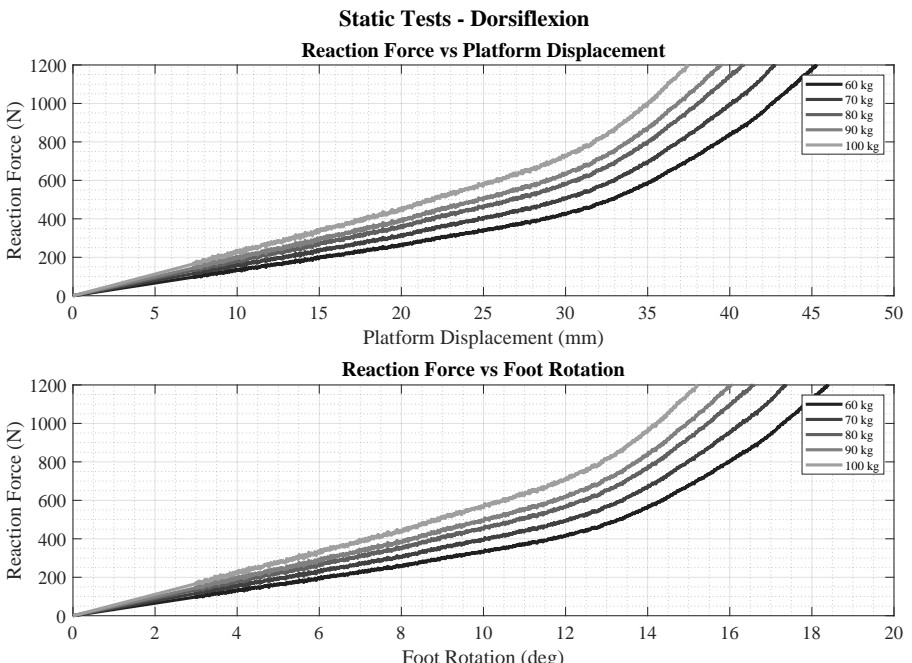

**Figure 8.** Dorsiflexion static tests given as reaction force vs. platform displacement (**top**) and reaction force vs. foot rotation (**bottom**) for the five prototypes of MyFlex-$\delta$.

The elastic energy for each foot prosthesis was calculated as the area below the curve through the trapezoid method up to the platform displacement which corresponds to a reaction force equivalent to the 108% of the weight category. The reaction force value for each of the weight categories is shown in Table 3, as well as the displacement of the platform, the corresponding rotation of the foot, and the elastic energy accumulated in absolute and normalized with respect to the weight category.

**Table 3.** Static tests results (dorsiflexion) and stored elastic energy.

|  | Weight Categories | | | | |
|---|---|---|---|---|---|
|  | 60 kg | 70 kg | 80 kg | 90 kg | 100 kg |
| Reaction force at 108% BW (N) | 636 | 742 | 848 | 954 | 1059 |
| Platform displ. at 108% BW (mm) | 36.04 | 35.83 | 35.79 | 36.18 | 35.79 |
| Rotation at 108% BW (°) | 14.64 | 14.55 | 14.54 | 14.69 | 14.54 |
| Stored energy at 108% BW (J) | 9.57 | 11.20 | 12.84 | 14.36 | 16.05 |
| Normalized stored energy at 108% BW (J/kg) | 0.16 | 0.16 | 0.16 | 0.16 | 0.16 |

The elastic energy accumulated by the MyFlex-$\delta$ prostheses, normalized for the weight categories, can be compared with the positive work of the human ankle during the push-off. The work of the ankle during the push-off is between 0.16 J/kg and 0.26 J/kg according to the results reported by Herr and Grabowski [14] and 0.21 J/kg and 0.37 J/kg according to the study conducted by Takahashi and Stanhope [114]. According to these two studies, the positive work by the human ankle increases as the walking speed increases.

### 3.2. Results from the Human Subjects Testing

In this section, the results from the human subjects testing are presented. In particular, the vertical ground reaction forces (Section 3.2.1), the ankle rotation in the sagittal plane, the plantarflexion at toe strike and the dorsiflexion at heel-off (Section 3.2.2), the ankle rotation in the frontal plane (Section 3.2.3), the step length (Section 3.2.4) and the stance

phase events and subphases duration are given (Section 3.2.5). All these measured and calculated parameters were obtained considering 50 consecutive gait cycles.

### 3.2.1. Ground Reaction Forces

The ground reaction forces were directly measured during the human subjects testing through force plates on the treadmill. The mean values of the ground reaction forces vs. the gait cycle are shown in Figure 9. For each leg, the mean values and the standard deviations of the maximum values of the ground reaction force were calculated from 50 gait cycles. Those maximum values with their mean values and standard deviations are reported in the first section of Table 4 ("Maximum Ground Reaction Forces") as absolute values, and normalized with respect to the body weight of the participants in the second section of the same Table ("Maximum Ground Reaction Forces/Body Weight/9.81 m/s$^2$").

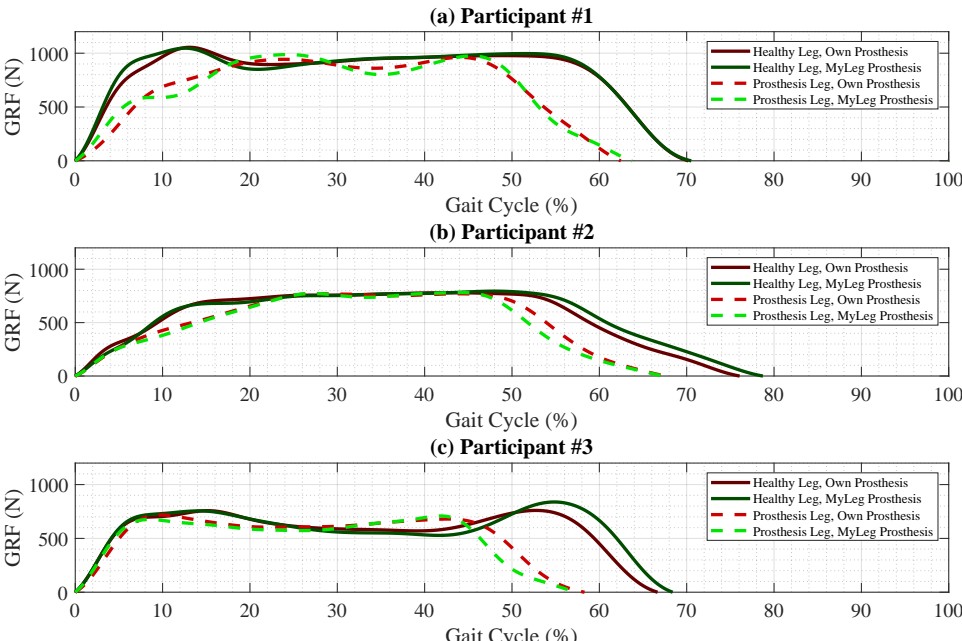

**Figure 9.** The ground reaction forces mean value calculated from 50 gait cycles of (**a**) Participant 1, (**b**) Participant 2 and (**c**) Participant 3: healthy leg, using their own prosthesis (dark red straight line); healthy leg, using MyFlex-$\delta$ (dark green straight line); prosthetic leg, using their own prosthesis (light red dashed line); prosthetic leg, using MyFlex-$\delta$ (light green dashed line).

To compare the ground reaction force values measured from the healthy foot and the prosthetic foot, the procedure illustrated in Figure 10a was followed and the results are reported in the third section of Table 4 ("Prosthetic Leg Ground Reaction Forces/Healthy Leg Ground Reaction Forces"). To compare the measured ground reaction force values while the participants were using their prosthesis and using MyFlex-$\delta$, the procedure illustrated in Figure 10b was followed, and the results are reported in the fourth section of Table 4 ("Ground Reaction Forces Using MyFlex-$\delta$/Ground Reaction Forces Using their Own Prosthesis").

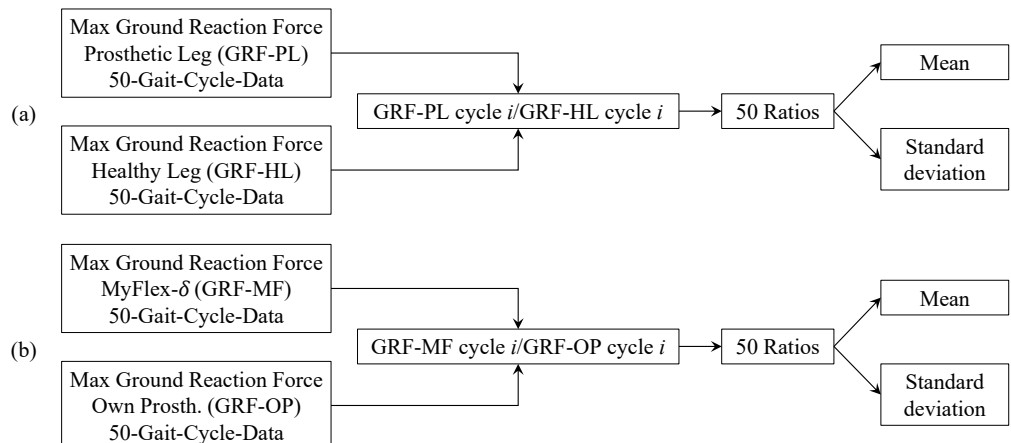

**Figure 10.** Procedures for the calculation of: (**a**) the ratio between the ground reaction force measured from the prosthetic leg (GRF-PL) and the healthy leg (GRF-HL); (**b**) the ratio between the ground reaction force measured while using MyFlex-$\delta$ (GRF-MF) and while using their own prosthesis (GRF-OP).

**Table 4.** Maximum ground reaction forces. Mean values (mean) and standard deviations (std) are calculated from 50 gait cycles.

| | Participant 1 | | Participant 2 | | Participant 3 | |
|---|---|---|---|---|---|---|
| **Maximum Ground Reaction Forces** | | | | | | |
| Leg and prosthesis used | mean | std | mean | std | mean | std |
| Healthy leg, own prosthesis | 1088 N | 37 N | 809 N | 6 N | 820 N | 31 N |
| Healthy leg, MyLeg prosthesis | 1078 N | 27 N | 824 N | 14 N | 872 N | 30 N |
| Prosthetic leg, own prosthesis | 993 N | 18 N | 808 N | 9 N | 760 N | 50 N |
| Prosthetic leg, MyLeg prosthesis | 1014 N | 19 N | 814 N | 9 N | 751 N | 25 N |
| **Maximum Ground Reaction Forces/Body Weight/9.81 m/s$^2$** | | | | | | |
| Leg and prosthesis used | mean | std | mean | std | mean | std |
| Healthy leg, own prosthesis | 107% | 4% | 103% | 1% | 114% | 4% |
| Healthy leg, MyLeg prosthesis | 106% | 3% | 104% | 2% | 121% | 4% |
| Prosthetic leg, own prosthesis | 98% | 2% | 102% | 1% | 106% | 7% |
| Prosthetic leg, MyLeg prosthesis | 100% | 2% | 103% | 1% | 104% | 3% |
| **Prosthetic Leg Ground Reaction Forces/Healthy Leg Ground Reaction Forces** | | | | | | |
| Prosthesis used | mean | std | mean | std | mean | std |
| Own prosthesis | 91.4% | 3.7% | 99.9% | 1.4% | 92.8% | 7.2% |
| MyFlex-$\delta$ | 94.0% | 3.0% | 98.8% | 1.6% | 86.3% | 4.5% |
| **Ground Reaction Forces Using MyFlex-$\delta$/Ground Reaction Forces Using their Own Prosthesis** | | | | | | |
| Leg side | mean | std | mean | std | mean | std |
| Healthy leg | 99.2% | 3.9% | 101.9% | 1.8% | 106.4% | 4.7% |
| Prosthetic leg | 102.1% | 2.9% | 100.7% | 1.9% | 99.1% | 6.8% |

### 3.2.2. Sagittal Plane Kinematics

Both the ankle rotations in the sagittal and frontal planes were calculated exploiting the positions and displacements of the markers. The rotations of the foot shown in Figure 11 were calculated as the difference between the ankle angle during the gait cycle and the ankle angle in the exact moment of the heel strike. This calculation was done in such a way that at the beginning of the gait cycle the rotation was $0°$. For each cycle, the plantarflexion at toe strike corresponded to the minimum value of the rotation of the ankle during the early stance, while the dorsiflexion at heel-off corresponded to the maximum value of the rotation of the ankle during the midstance. For each leg, the mean values and the standard deviations of the plantarflexion at toe strike and the dorsiflexion at heel-off were calculated from 50 gait cycles. These parameters with their mean values and standard deviations are reported in the first sections of Table 5 ("hlPlantarflexion at Toe Strike") and Table 6 (hl"Dorsiflexion at Heel-Off"). To compare the plantarflexion at toe strike and the dorsiflexion at heel-off measured and calculated from the healthy foot and the prosthetic foot, the procedure illustrated in Figure 12a was followed and the results are reported in the second section of Table 5 ("Prosthetic Leg Plantarflexion–Healthy Leg Plantarflexion") and second section of Table 6 ("Prosthetic Leg Dorsiflexion–Healthy Leg Dorsiflexion"). To compare the plantarflexion at toe strike and the dorsiflexion at heel-off measured and calculated while the participants were using their prosthesis and using MyFlex-$\delta$, the procedure illustrated in Figure 12b was followed, and the results are reported in the third section of Table 5 ("Plantarflexion Using MyFlex-$\delta$–Plantarflexion Using their Own Prosthesis") and third section of Table 6 ("Dorsiflexion Using MyFlex-$\delta$ –Dorsiflexion Using their Own Prosthesis").

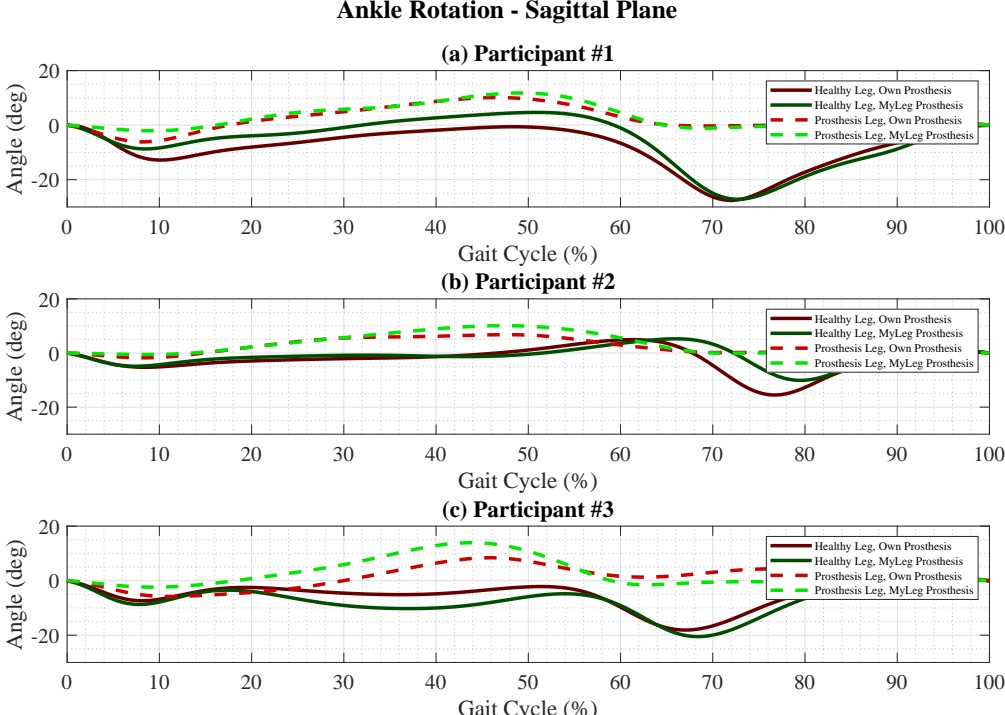

**Figure 11.** The ankle dorsiflexion($+$) and plantarflexion($-$) during gait cycle of (**a**) Participant 1, (**b**) Participant 2 and (**c**) Participant 3: healthy leg, using their own prosthesis (dark red straight line); healthy leg, using MyFlex-$\delta$ (dark green straight line); prosthetic leg, using their own prosthesis (light red dashed line); prosthetic leg, using MyFlex-$\delta$ (light green dashed line).

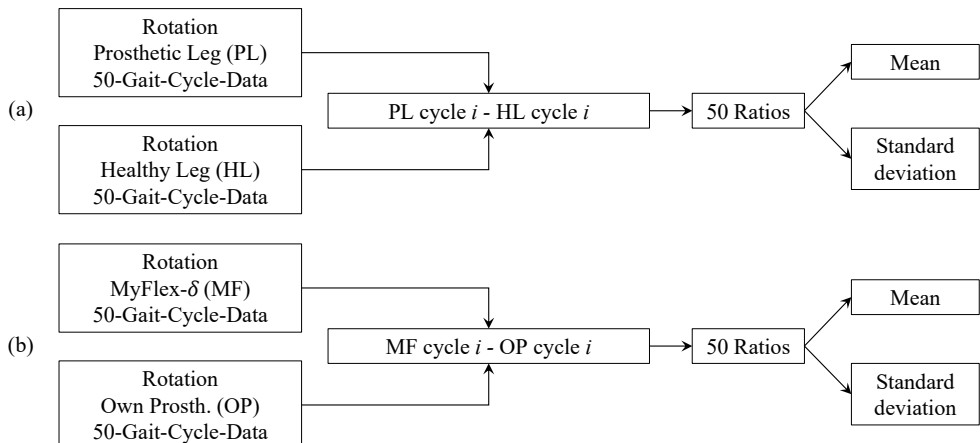

**Figure 12.** Procedures for the calculation of: (**a**) the difference between the rotation measured from the prosthetic leg and the healthy leg; (**b**) the difference between the rotation measured while using MyFlex-$\delta$ and while using their own prosthesis.

**Table 5.** Plantarflexion at toe strike. Mean values (mean) and standard deviations (std) are calculated from 50 gait cycles.

| | Participant 1 | | Participant 2 | | Participant 3 | |
|---|---|---|---|---|---|---|
| Plantarflexion at Toe Strike | | | | | | |
| Leg and prosthesis used | mean | std | mean | std | mean | std |
| Healthy leg, own prosthesis | 13.0° | 1.2° | 5.8° | 1.0° | 8.0° | 1.0° |
| Healthy leg, MyLeg prosthesis | 8.8° | 1.4° | 5.3° | 1.1° | 9.1° | 1.8° |
| Prosthetic leg, own prosthesis | 6.3° | 0.7° | 1.8° | 0.2° | 6.0° | 0.6° |
| Prosthetic leg, MyLeg prosthesis | 2.1° | 0.1° | 0.5° | 0.1° | 2.4° | 0.4° |
| Prosthetic Leg Plantarflexion–Healthy Leg Plantarflexion | | | | | | |
| Prothesis used | mean | std | mean | std | mean | std |
| Own prosthesis | −6.8° | 1.2° | −3.9° | 1.0° | −2.0° | 1.3° |
| MyFlex-$\delta$ | −6.7° | 1.5° | −4.8° | 1.1° | −6.7° | 2.0° |
| Plantarflexion Using MyFlex-$\delta$–Plantarflexion Using their Own Prosthesis | | | | | | |
| Leg side used | mean | std | mean | std | mean | std |
| Healthy Leg | −4.3° | 0.4° | −0.4° | 1.4° | +1.0° | 1.9° |
| Prosthetic Leg | −4.2° | 0.7° | −1.3° | 0.3° | −3.6° | 0.7° |

**Table 6.** Dorsiflexion at heel-off. Mean values (mean) and standard deviations (std) are calculated from 50 gait cycles.

| | Participant 1 | | Participant 2 | | Participant 3 | |
|---|---|---|---|---|---|---|
| Dorsiflexion at Heel-Off | | | | | | |
| Leg and prosthesis used | mean | std | mean | std | mean | std |
| Healthy leg, own prosthesis | −0.8° | 1.7° | 5.0° | 1.8° | −1.8° | 2.9° |
| Healthy leg, MyLeg prosthesis | 4.7° | 1.4° | 4.6° | 1.9° | −3.7° | 2.0° |
| Prosthetic leg, own prosthesis | 10.2° | 0.4° | 6.9° | 0.2° | 8.3 | 0.8° |
| Prosthetic leg, MyLeg prosthesis | 11.9° | 0.3° | 10.2° | 0.3° | 14.0° | 0.5° |

**Table 6.** *Cont.*

| | Participant 1 | | Participant 2 | | Participant 3 | |
|---|---|---|---|---|---|---|
| Prosthetic Leg Dorsiflexion–Healthy Leg Dorsiflexion | | | | | | |
| Prosthesis used | mean | std | mean | std | mean | std |
| Own prosthesis | +11.0° | 1.7° | +1.8° | 1.7° | +10.0° | 3.0° |
| MyFlex-$\delta$ | +7.1° | 1.5° | +5.6° | 2.0° | +17.7° | 2.0° |
| Dorsiflexion Using MyFlex-$\delta$–Dorsiflexion Using their Own Prosthesis | | | | | | |
| Leg side | mean | std | mean | std | mean | std |
| Healthy Leg | +5.5° | 2.3° | −0.5° | 2.8° | −2.0° | 2.6° |
| Prosthetic Leg | +1.6° | 0.5° | +3.3° | 0.4° | +5.7° | 1.0° |

### 3.2.3. Frontal Plane Rotation

Activities during familiarization were neither measured kinetically nor kinematically. The measurements were carried out only in the final test on the treadmill, through force plates (ground reaction force measurement) and through position markers (kinematics measurement of each segment and subsegment of the lower limb and torso). Therefore, the evaluations of the various activities such as walking sideways (Figure 4b), turning step after walking straight on level ground (Figure 4c), walking on an inclined and declined treadmill (Figure 4d), and walking on uneven terrain (Figure 4e), were done in a qualitative manner and reported in the following sections as comments from the participants. During familiarization activities with MyFlex-$\delta$, the extra degrees of freedom were specifically and positively noted by the three participants, who gave positive feedback on their perceptions of comfort and foot adaptability to the relative angle between the ground and the leg, even in the case of a completely horizontal floor, and not only on uneven ground. In fact, as has been seen in other studies on level walking [49] or also on the measurements of the kinematics of the healthy leg of the three participants (Figure 13, healthy legs), the foot also rotates in the frontal plane. This rotation in the frontal plane is partly due to the transverse distance between the foot and the axis of the human body.

Observing Figure 14, even if represented in an exaggerated way, as a variation of the transverse distance brings a rotation of the foot in the frontal plane, a greater distance generates an inversion of the foot around the ankle, while a smaller distance generates an eversion. This frontal rotation is necessary to keep the foot flat to the ground. In addition, during the walk, the trunk oscillates in the frontal plane [115], reducing or increasing the distance in the frontal plane between the axis of the human body and the foot during a single gait cycle, also generating in this case a need for the foot to rotate around the ankle in eversion and inversion to keep the foot flat to the ground. Figure 13 displays the rotation of the foot in the frontal plane during the test with the treadmill with sensors (force plates and markers), where participants were asked to walk on a horizontal plane at a comfortable speed to them. Observing the dashed curves of Figure 13, which describe the rotations in the frontal plane of the prosthetic foot with respect to the ankle, it can be seen how the light green dashed curves (Myflex-$\delta$) give greater rotations than the light red dashed curves (their prostheses). This means that MyFlex-$\delta$ provides more flexibility in the frontal plane than their prostheses. The positive feelings perceived by participants during other activities during the familiarization (Figure 4) may be due to this property of Myflex-$\delta$ to have flexibility even in the frontal plane.

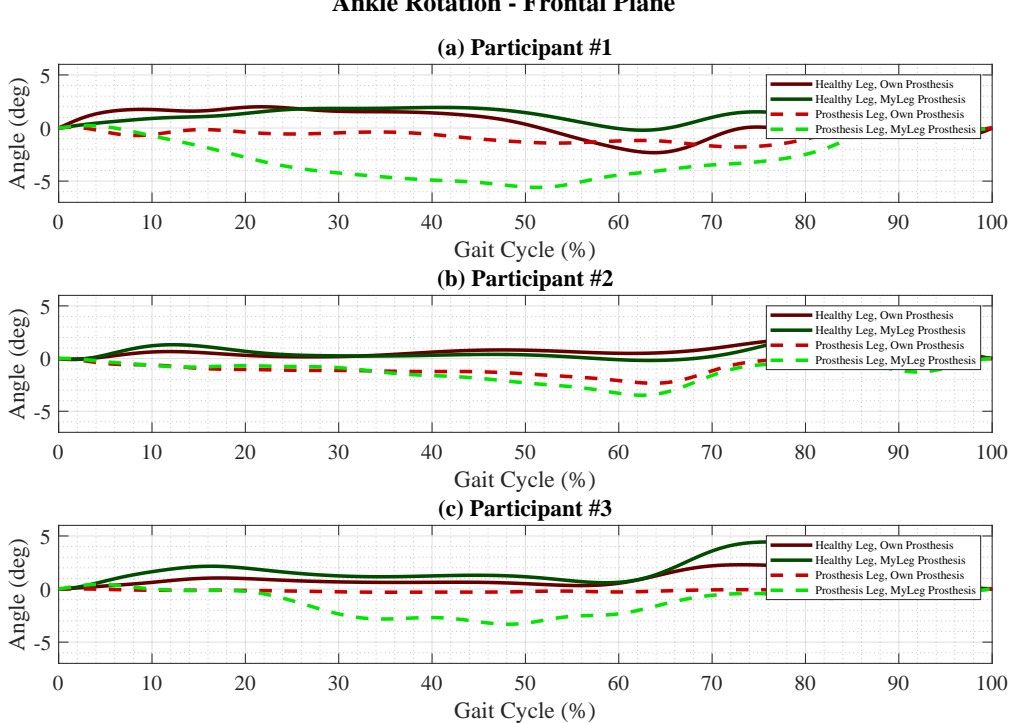

**Figure 13.** The eversion(+) and inversion(−) during gait cycle of (**a**) Participant 1, (**b**) Participant 2 and (**c**) Participant 3: healthy leg, using their own prosthesis (dark red straight line); healthy leg, using MyFlex-$\delta$ (light red straight line); prosthetic leg, using their own prosthesis (dark red dashed line); prosthetic leg, using MyFlex-$\delta$ (light red dashed line).

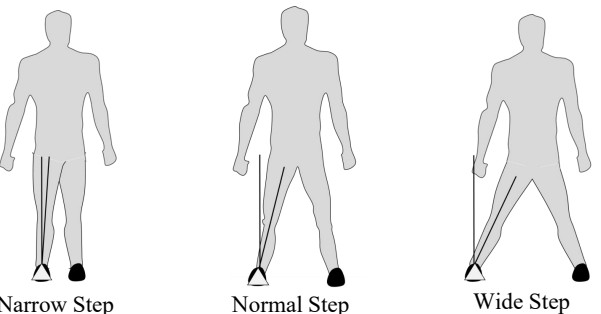

**Figure 14.** The variation of the angle of the ankle in the front plane depending on the step width.

### 3.2.4. Step Length

The prosthetic leg stiffness has a direct influence on the step length [71,72,116,117]. Therefore, in this paper the step length was considered as a parameter to assess gait symmetry. A prosthetic foot provides the right stiffness and, therefore, improves the symmetry of the walk if it reduces the difference between the length of the step made with the healthy leg and that made with the prosthesis. The step length of each leg was calculated considering the treadmill velocity and the center of pressure positions of the foot. The mean values and standard deviations of the step length for each leg and for each participant are reported in the first section of Table 7 (Step Length).

To compare the step length measured and calculated from the healthy foot and the prosthetic foot, the procedure illustrated in Figure 15a was followed and the results are reported in the second section of Table 7 ("Prosthetic Leg Step Length/Healthy Leg Step Length"). To compare the step length measured and calculated while the participants were using their prosthesis and using MyFlex-$\delta$, the procedure illustrated in Figure 15b was

followed, and the results are reported in the fourth section of Table 7 ("Step length Using MyFlex-$\delta$/Step Length Using their Own Prosthesis").

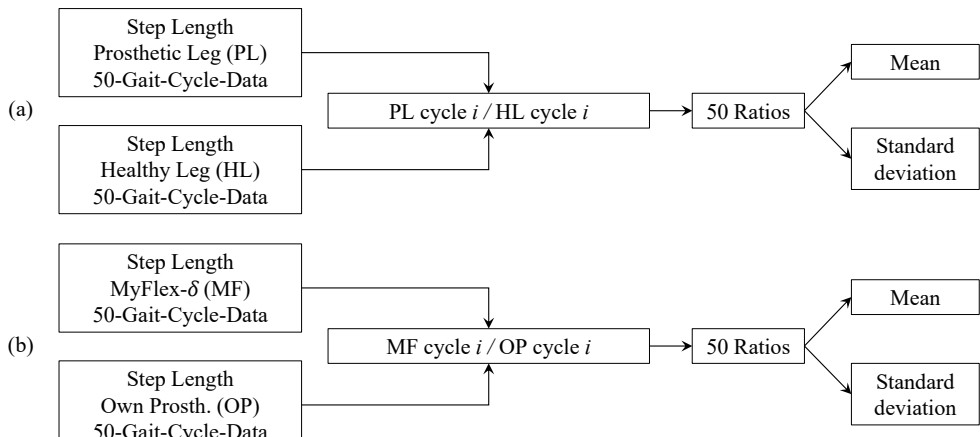

**Figure 15.** Procedures for the calculation of: (**a**) the ratio between the step length measured from the prosthetic leg and the healthy leg; (**b**) the ratio between the step length measured while using MyFlex-$\delta$ and while using a participant's own prosthesis.

**Table 7.** Step Length. Mean values (mean) and standard deviations (std) are calculated from 50 gait cycles.

| | Participant 1 | | Participant 2 | | Participant 3 | |
|---|---|---|---|---|---|---|
| | **Step Length** | | | | | |
| Leg and prosthesis used | mean | std | mean | std | mean | std |
| Healthy leg, own prosthesis | 0.534 m | 0.014 m | 0.357 m | 0.027 m | 0.554 m | 0.046 m |
| Healthy leg, MyFlex-$\delta$ | 0.535 m | 0.015 m | 0.358 m | 0.020 m | 0.562 m | 0.018 m |
| Prosthetic leg, own prosthesis | 0.478 m | 0.022 m | 0.256 m | 0.019 m | 0.646 m | 0.030 m |
| Prosthetic leg, MyFlex-$\delta$ | 0.471 m | 0.015 m | 0.242 m | 0.018 m | 0.624 m | 0.016 m |
| | **Prosthetic Leg Ankle Range of Motion/Healthy Leg Step Length** | | | | | |
| Prosthesis used | mean | std | mean | std | mean | std |
| Own prosthesis | 89.3% | 4.8% | 72.0% | 7.9% | 117.9% | 17.3% |
| MyFlex-$\delta$ | 88.1% | 3.2% | 67.7% | 6.6% | 111.2% | 4.7% |
| | **Step Length Using MyFlex-$\delta$/Using their Own Prosthesis** | | | | | |
| Leg side | mean | std | mean | std | mean | std |
| Healthy leg | 100.2% | 3.7% | 100.9% | 10.0% | 102.5% | 15.3% |
| Prosthetic leg | 99.0% | 5.6% | 94.9% | 9.5% | 96.7% | 4.6% |

### 3.2.5. Stance Phase

The properties of prosthetic feet must promote early flat foot [97,116]: this means that, the shorter the duration of the early stance, the sooner the flat foot phase begins. In order to calculate the length of the early stance, it is necessary to determine when the toe strike occurs. In fact, the early stance begins at 0% of the gait cycle and ends when the toe strike occurs. From the kinematic data, it is possible to determine when the toe strike occurs by determining when the inversion of the rotation of the foot around the ankle occurs, from plantarflexion to dorsiflexion. To understand how long the flat foot phase lasts, and therefore how long the stage in which the prosthetic foot performs a rotation of dorsiflexion accumulating elastic energy with its elastic elements, it is necessary to determine when the flat foot ends, i.e., when the heel-off takes place. As was the case for the determination of the end of the early stance (toe strike), the end of the midstance (heel-off) was found by determining the change of rotation of the foot from dorsiflexion to plantarflexion. Once

again, the kinematic data in the sagittal plane were exploited. The end of the stance phase (toe-off) was determined in two ways: by calculating the ratio of contact time (parameter measured during human subjects testing) to stride time (also measured), or by determining the moment when the ground reaction force becomes zero (which means that the foot is no longer in contact with the ground). Toe strike, heel-off and toe-off events were calculated for each cycle. Once calculated, mean values and standard deviations were calculated for each of them. The calculated values are given in Table 8.

Knowing that the early stance starts at 0% of the gait cycle and ends when the toe strike occurs, and already knowing when the toe strike occurs, it is easy to deduce how to calculate the duration of the early stance. Knowing for each cycle when the toe strike occurs and when the heel-off occurs, it is easy to also calculate the duration of the midstance. For each cycle, as well as for the early stance, the duration of the midstance was calculated, and subsequently mean values and standard deviations were calculated. The same procedure was followed for the calculation of the late stance, knowing for each cycle the end of the stance phase. The durations of the subphases of the stance phase are given in Table 9.

**Table 8.** Stance phase events. Mean values (mean) and standard deviations (std) are calculated from 50 gait cycles.

|  | Participant 1 | | Participant 2 | | Participant 3 | |
| --- | --- | --- | --- | --- | --- | --- |
| **Toe Strike** | | | | | | |
| Leg and prosthesis used | mean | std | mean | std | mean | std |
| Healthy leg, own prosthesis | 10.3% | 0.6% | 9.3% | 1.9% | 8.5% | 0.6% |
| Healthy leg, MyFlex-$\delta$ | 8.6% | 0.5% | 8.0% | 1.2% | 8.0% | 0.5% |
| Prosthetic leg, own prosthesis | 8.4% | 0.8% | 8.4% | 0.9% | 11.2% | 0.7% |
| Prosthetic leg, MyFlex-$\delta$ | 9.3% | 0.5% | 8.6% | 1.5% | 9.6% | 0.9% |
| **Heel-Off** | | | | | | |
| Leg and prosthesis used | mean | std | mean | std | mean | std |
| Healthy leg, own prosthesis | 48.5% | 1.8% | 61.4% | 2.1% | 51.4% | 1.8% |
| Healthy leg, MyFlex-$\delta$ | 51.0% | 1.3% | 66.1% | 1.7% | 54.2% | 1.0% |
| Prosthetic leg, own prosthesis | 46.9% | 0.9% | 48.3% | 1.6% | 45.7% | 1.0% |
| Prosthetic leg, MyFlex-$\delta$ | 49.4% | 1.0% | 47.4% | 1.6% | 43.9% | 0.9% |
| **Toe-Off** | | | | | | |
| Leg and prosthesis used | mean | std | mean | std | mean | std |
| Healthy leg, own prosthesis | 74.3% | 7.4% | 77.6% | 1.8% | 67.1% | 1.6% |
| Healthy leg, MyFlex-$\delta$ | 73.6% | 2.2% | 79.1% | 2.0% | 70.6% | 5.0% |
| Prosthetic leg, own prosthesis | 63.3% | 1.0% | 69.2% | 1.4% | 61.0% | 7.3% |
| Prosthetic leg, MyFlex-$\delta$ | 64.2% | 1.3% | 68.6% | 1.7% | 58.4% | 1.0% |

**Table 9.** Stance phase subphases durations. Mean values (mean) and standard deviations (std) are calculated from 50 gait cycles.

|  | Participant 1 | | Participant 2 | | Participant 3 | |
| --- | --- | --- | --- | --- | --- | --- |
| **Early Stance Duration** | | | | | | |
| Leg and prosthesis used | mean | std | mean | std | mean | std |
| Healthy leg, own prosthesis | 10.3% | 0.6% | 9.3% | 1.9% | 8.5% | 0.6% |
| Healthy leg, MyFlex-$\delta$ | 8.6% | 0.5% | 8.0% | 1.2% | 8.0% | 0.5% |
| Prosthetic leg, own prosthesis | 8.4% | 0.8% | 8.4% | 0.9% | 11.2% | 0.7% |
| Prosthetic leg, MyFlex-$\delta$ | 9.3% | 0.5% | 8.6% | 1.5% | 9.6% | 0.9% |

**Table 9.** *Cont.*

|  | Participant 1 | | Participant 2 | | Participant 3 | |
|---|---|---|---|---|---|---|
| | Mid Stance Duration | | | | | |
| Leg and prosthesis used | mean | std | mean | std | mean | std |
| Healthy leg, own prosthesis | 38.2% | 1.8% | 52.1% | 2.9% | 42.3% | 3.1% |
| Healthy leg, MyFlex-$\delta$ | 42.4% | 1.4% | 58.1% | 2.1% | 45.6% | 3.1% |
| Prosthetic leg, own prosthesis | 38.5% | 1.1% | 39.9% | 1.8% | 34.5% | 1.4% |
| Prosthetic leg, MyFlex-$\delta$ | 40.1% | 0.9% | 38.9% | 2.2% | 34.3% | 1.0% |
| | Late Stance Duration | | | | | |
| Leg and prosthesis used | mean | std | mean | std | mean | std |
| Healthy leg, own prosthesis | 25.8% | 6.6% | 16.2% | 1.7% | 15.7% | 1.4% |
| Healthy leg, MyFlex-$\delta$ | 22.6% | 1.9% | 13.6% | 1.8% | 16.4% | 5.0% |
| Prosthetic leg, own prosthesis | 16.4% | 0.8% | 20.9% | 1.4% | 15.4% | 7.2% |
| Prosthetic leg, MyFlex-$\delta$ | 14.8% | 0.6% | 21.2% | 1.0% | 14.5% | 0.4% |

## 4. Discussion

Once certified, clinical tests with the three participants having a transfemoral amputation were performed. The three participants had different weights, but still within the range for which the five MyFlex-$\delta$ had been optimized. The three participants were able to try at least two different MyFlex-$\delta$ stiffnesses and choose the most suitable one, according to their sensations. All participants had a relatively short time (4 h) to choose and familiarize themselves with the chosen prosthesis. In addition, the three participants had a transfemoral amputation and wore powered knee prostheses set for the characteristics and behavior of their foot prosthesis and not for that of Myflex-$\delta$. Following the same test protocol used to test MyFlex-$\delta$, the participants also tested in the laboratory the prostheses they use daily. The results were then compared.

### 4.1. MyFlex-$\delta$ vs. Their Own Prosthesis

In addition, because of the lack of experience that participants had with Myflex-$\delta$, there were no major improvements from the perspective of gait symmetry. However, almost all the evaluation parameters chosen by the authors had a difference far below the 10%, initially taken as a minimum target.

#### 4.1.1. Ground Reaction Forces

Silverman and Neptune [118] reviewed from other works that unilateral transtibial amputation results in the loss of vital muscles for body support, propulsion, midlateral balance, and swing phase initiation [119–123]. This loss generates less symmetry in walking, and causes leg and back pain [124,125]. According to some studies, amputees present more ground reaction force with the healthy leg [126–128]. The same results were obtained in this work with the three participants, as shown in the second section of Table 4 ("Prosthetic Leg Ground Reaction Force/Healthy Leg Ground Reaction Force"). The smaller the difference between the maximum values of the ground reaction force of the healthy leg and the amputated leg, the better for the symmetry of the gait. In other words, the closer the ratio between the ground reaction force given by the prosthetic leg and the ground reaction force given by the healthy leg to 100%, the greater the gait symmetry. As shown in the third section of Table 4 ("Ground Reaction Forces Using MyFlex-$\delta$/Ground Reaction Forces Using their Own Prosthesis"), current results of the MyFlex-$\delta$ on ground reaction force symmetry are contradicting, as one participant improved their symmetry (Participant 1, going from a ratio of 91.4% to 94.0%), when using MyFlex-$\delta$ instead of using their own prosthesis, as shown in the second section of Table 4), one participant showed similar ratios (Participant 2, from 99.9% to 98.8%) and one showed less symmetry (Participant 3, from 93% to 86%).

### 4.1.2. Sagittal Plane Kinematics

*Plantarflexion at Toe Strike: Prosthetic Leg vs. Healthy Leg*

The second section of Table 5 ("Prosthetic Leg Plantarflexion–Healthy Leg Plantarflexion") shows how, regardless of the prosthesis used, the plantarflexion with the prosthetic foot is less than that with the healthy foot.

*Plantarflexion at Toe Strike: MyFlex-δ vs. Own Prosthesis*

According to their feedback, the three participants perceived a lower absorption of the impact between the foot and the ground during the heel strike, while using MyFlex-δ. Unfortunately, this was probably due to a problem of interference between two components during the plantarflexion (Figure 7). This perception can be confirmed by the diminished plantarflexion at the end of the early stance of the prosthetic leg, while using MyFlex-δ, compared to the plantarflexion of their own prosthesis. Indeed, observing the results given in the third section of Table 5 ("Plantarflexion Using MyFlex-δ–Plantarflexion Using their Own Prosthesis"), in particular those concerning the prosthetic leg, Myflex-δ gave less plantarflexion than the participant's own prosthesis.

*Dorsiflexion at Heel-Off: Prosthetic Leg vs. Healthy Leg*

In the second section ("Prosthetic Leg Dorsiflexion–Healthy Leg Dorsiflexion") of Table 6 the differences between the dorsiflexion measured at the prosthetic leg and the dorsiflexion measured at the healthy leg, both for a participant's prosthesis and for MyFlex-δ, were reported. From the results, it is clear that the dorsiflexion for both prostheses is greater in the prosthetic foot.

*Dorsiflexion at Heel-Off: MyFlex-δ vs. Own Prosthesis*

As can be seen in the second section of Table 6 ("Dorsiflexion Using MyFlex-δ–Dorsiflexion Using their Own Prosthesis"), for all three participants, from their own prosthesis to MyFlex-δ, there was an increase in dorsiflexion with the prosthetic leg, which could be equivalent to the increased elastic energy stored by the elastic elements of the prosthesis.

### 4.1.3. Step Length

A prosthetic foot has the desired stiffness, and therefore guarantees greater gait symmetry when the length of the step with the prosthetic leg does not differ much from the length of the step with the healthy leg [71,72,116,117].

*Prosthetic Leg vs. Healthy Leg*

As for the comparison between the healthy leg and the prosthetic leg, there were contradictory results, as the prosthetic leg step was shorter than that of the healthy leg for the first two participants, while for the third it was the opposite (Table 7, section: "Prosthetic Leg Ankle Range of Motion/Healthy Leg Step Length"). This result was independent of the prosthesis used.

*MyFlex-δ vs. Own Prosthesis*

As for the step length, Myflex-δ had negative outcomes compared to their prosthesis for the first and second participants, while for the third participant there was an improvement since the ratio reported in the second section of Table 7 ("Prosthetic Leg Ankle Range of Motion/Healthy Leg Step Length") was closer to 100% when Participant 3 used their own prosthesis (from 117.9% with their own prosthesis to 111.2% with MyFlex-δ). The worsening for the first two participants, however, did not go beyond 5% (from 89.3% to 88.1% for Participant 1 and 72.0% to 67.7% for Participant 2). In the third section of Table 7 ("Step Length Using MyFlex-δ/Using their Own Prosthesis"), we saw how the ratio between the step length made with the prosthetic leg using MyFlex-δ and the step length made with the healthy leg using a participant's prosthesis was always less than 100%: this means that all three participants took a longer step with the prosthetic leg when using their prosthesis

than when using Myflex-$\delta$. The difference, however, was again below 10%; indeed, the ratios were: 99.0% (Participant 1), 94.9% (Participant 2) and 96.7% (Participant 3).

### 4.1.4. Stance Phase

Early Stance

The properties of prosthetic feet must promote early flat foot [97,116]: this means that, the shorter the duration of the early stance, the sooner the flat foot phase begins. Based on this statement, and comparing the two prostheses through the values calculated and reported in the first section of Table 9, for the first participant, the early stance of the prosthetic foot lasted longer when using MyFlex-$\delta$ (from 8.4% with their own prosthesis to 9.3% with MyFlex-$\delta$), while for the second participant the values were almost the same (from 8.4% to 8.6%), and for the third participant there was an improvement with the prosthesis proposed in this paper (from 11.2% to 9.6%).

Midstance

The midstance phase is the stage where the foot is in flat foot. According to the authors' knowledge, no information has been reported on the importance of the duration of the midstance. However, this step corresponds to the phase in which the prosthesis stores the elastic energy needed for the push-off. Looking only at the data concerning the prosthetic legs in Table 9, the values are more or less the same, with a slightly greater difference for the first participant (40.1% of the duration of the midstance with MyFlex-$\delta$ vs. 38.5% with their prosthesis), while the other two values are more or less the same (38.9% vs. 39.9% for the second participant and 34.3% vs. 34.5% for the third). Comparing, instead, the values of the duration of the midstance of the prosthetic leg, regardless of the prosthesis used, the duration is shorter than the duration of the midstance of the healthy leg, with differences above 10% mainly for the second and third participant.

Late Stance

For prostheses, especially for ESR prostheses, the duration of the late stance phase (or push-off phase) corresponds to the duration with which the elastic energy stored during the midstance is released. The first and third participants, especially the first one, pointed out that the elastic energy accumulated by MyFlex-$\delta$ was released faster than that accumulated by their prosthesis. For the second participant, however, it was not the same. This feedback was confirmed by the data reported in Table 9 regarding the duration of the late stance, if those concerning the two prostheses are compared. In fact, as reported in Table 9, the first participant recorded an average value of the duration of the late stance of 14.5% with MyFlex-$\delta$ against 22.6% with their own prosthesis, while Participant 2 values were 21.2% with MyFlex-$\delta$ and 13.6% with their prosthesis. Finally, the third participant recorded a duration in percentage of 14.5% of the late stance with the prosthesis proposed by the authors against 16.4% of the late stance with their prosthesis.

### 4.2. Participants' Feedback on MyFlex-$\delta$

As hypothesized, the addition of the spherical joint did enlarge angles at least in dorsiflexion and inversion/eversion. The extra degrees of freedom of the MyFlex-$\delta$ might help adapt to uneven terrain while creating stability. Only plantarflexion angles were diminished when using the MyFlex-$\delta$. This was noticed by the participants through a hard impact during heel strikes. As previously mentioned, this was probably caused by the interference between the foot shell/shoe and the middle blade, as show in Figure 7, which created a hard end stop. When evaluating these results, it should be taken into account that the familiarization period was rather short, so accordingly, participants were likely not fully adapted to walking with the MyFlex-$\delta$. However, results and feedback from the participants do indicate the addition of a spherical joint to an ESR foot prosthesis can provide extra degrees of freedom during gait.

## 5. Conclusions

In this paper, the authors presented the results of preliminary human subject testing performed on Myflex-$\delta$, an ESR foot prosthesis with a spherical joint at the ankle. First, the main features of the prosthesis were briefly described: the spherical ankle joint, combined with the elastic elements of the ESR foot, provided flexibility in all directions of rotation. Second, the optimization of five Myflex-$\delta$ examples with five different stiffnesses for as many weight categories of users was briefly presented. Third, human subject tests with three patients of three different weights were presented. Finally, the results were shown and discussed. The three participants had a relatively short time to familiarize themselves with Myflex-$\delta$. In fact, the results showed no substantial improvement from the point of view of gait symmetry, when comparing the results obtained with Myflex-$\delta$ with those obtained with participants' own prostheses. However, the addition of a spherical ankle joint, gathered better perceptions in activities such as turning step or walking on uneven terrain. In addition, Myflex-$\delta$ examples tested by the three patients showed a greater range of motion in the sagittal plane, which may mean more elastic energy accumulated by the ESR foot. Future work will involve tests with a larger number of participants, always included in the weight range between 60 and 100 kg and with a K3 or K4 ambulation level. In this way, conclusions regarding the advantages of the spherical ankle can be drawn.

**Author Contributions:** Conceptualization, J.T., V.G.M.K., R.C. and A.Z.; methodology, V.G.M.K., J.T., R.A.L., M.L., T.M.B., G.T., V.R., E.S. and P.B.; validation, J.T., V.G.M.K., R.A.L., M.L., T.M.B., G.T., V.R., E.S. and P.B.; formal analysis, J.T. and V.G.M.K.; investigation, V.G.M.K., J.T., R.A.L., G.T., V.R. and E.S.; data curation, V.G.M.K. and J.T.; writing—original draft preparation, J.T. and V.G.M.K.; writing—review and editing, J.T., V.G.M.K., R.A.L., N.V., M.L., T.M.B., G.T., V.R., E.S., P.B., M.O., R.C. and A.Z.; visualization, J.T., V.G.M.K. and A.Z.; supervision, M.O., N.V., R.C. and A.Z.; resources: M.O., A.Z. and N.V.; project administration, M.O., N.V., R.C. and A.Z.; funding acquisition, R.C. and A.Z. All authors have read and agreed to the published version of the manuscript.

**Funding:** This work was funded by the European Commission's Horizon 2020 Programme as part of the project MyLeg under grant no. 780871.

**Institutional Review Board Statement:** The study procedures were approved by the ethical committee CMO Arnhem-Nijmegen (2019–5920) and complied with the guidelines defined in the Declaration of Helsinki. The approval date of the ethical committee was 16 January 2020.

**Informed Consent Statement:** Informed consent was obtained from all subjects involved in the study.

**Data Availability Statement:** Not applicable.

**Acknowledgments:** The authors want to thank Stefano Monti (Department of Industrial Engineering technician, University of Bologna), Mauro and Lorenzo Sassatelli (Metal TIG S.R.l.—www.metaltig.it, accessed on 2 January 2022, Italy) for their contributions to the realization of the prototypes and Marco Papenburg (prosthetist, Papenburg Orthopedie, The Netherlands).

**Conflicts of Interest:** The authors declare no conflict of interest. The funders had no role in the design of the study; in the collection, analyses, or interpretation of data; in the writing of the manuscript, or in the decision to publish the results.

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
