# Peer review of "The Functionality Verification through Pilot Human Subject Testing of MyFlex-δ: An ESR Foot Prosthesis with Spherical Ankle Joint"

_applsci, doi:10.3390/app12094575_

Round 1

Reviewer 1 Report

The study evaluated a previously developed foot prosthesis with 3 human subjects who had suffered transfemoral amputation. The manuscript was well written and the results were clearly presented. Research topic is appropriate for the general readership of the journal. No major concerns were realized by this reviewer. However, some minor issues are suggested to be addressed by the authors.

(1) This reviewer can appreciate that the current study was based on earlier studies by the team that developed a foot prosthesis with spherical ankle joint, namely the MyFlex. While the authors referenced the previous work, it was not clear how the prosthesis was selected for the current study. For example, how did the authors select the 5 stiffnesses for the 3 subjects? How were the 5 stiffnesses developed in the first place? Were they based on real ankle stiffness data from other human subjects? Please provide some details.

(2) For the 3D FE model used in the study to help develop the prosthesis, how was the validation work conducted? For example, what about the sensitivity, accuracy, repeatability etc. of the FE model?

(3) As mentioned in the manuscript, with only 3 subjects being included in the study, no stats were carried out. The small sample size can be a limitation of the study. Please discuss. What if more subjects would have been available? Then, what would be the hypothesis of the study in that case?

(4) Based on the current results from the 3 subjects, what will the authors plan for future work (from themselves or others)?

Author Response

Dear Sir/Ma'am

please attached you can find our responses.

Thank you very much for reviewing our manuscript

Reviewer 2 Report

Overall feedback

The study entitled “The Functionality Verification through Pilot Human Subject Testings of MyFlex-δ : an ESR Foot Prosthesis with Spherical Ankle Joint” presents a new passive Energy-Storing-and-Releasing (ESR) foot, which differs from the version MyFlex-δ  for having a spherical ankle joint. This new feature has been introduced to allow foot adaptation to different ground conditions, especially on irregular or laterally inclined ground. The new prosthetic foot has been first described, ISO 10328-Equivalent static tests have been performed, and then tested on three K4 participants (year of amputation: 10+ years). The participants tried the prosthesis for one day using their prosthetic knee. To evaluate the performance of the proposed prosthetic foot, different parameters were explored: maximum ground reaction forces, plantarflexion at toe strike, dorsiflexion at heel off, ankle rotation from toe-strike to heel-off, step length, and stance/early stance/late stance durations. Overall, from a quantitative point of view, it was not observed an overall trend among the participant suggesting that the proposed foot is better than the one they are currently using. This was expected due limited time that the participants had to familiarise themselves with the newly proposed solution. Nevertheless, better “sensations” were overall reported during activities requiring a larger range of motion in the frontal and coronal planes.

Specific comments

The authors should include relevant references when addressing previous work. For example, page 2, lines 23-28; page 3, lines 86-90.

The authors present different multiaxial prosthetic feet available either on the market (i.e., “In several patents, to create multiaxial flexibility at the ankle joint, spherical joints have been used in conventional feet using elastomeric bumpers [17–21] or springs [22–24] to create rotational stiffness and damping in at least two rotations.”) or described “academically” (in section 1.1.2). To reinforce the strength of this study, the authors should stress how MyFlex-δ  differs from these solutions in the discussion section.

In section 2.4.2 the authors should:

  • State whether the motion capture system and the force platforms were synchronised;
  • Indicate the different parameters extracted during the analysis (maximum ground reaction forces, plantarflexion at toe strike, dorsiflexion at heel off, ankle rotation from toe-strike to heel-off, step length, and stance/early stance/late stance durations). It is important to specify here how those parameters were calculated.

In the whole manuscript, the authors refer to either symmetry or asymmetry. I would suggest always referring to one of the two.

The main quantitative results are reported in different figures (Figs 7,8,10,12,13,16,17,18,19, and 20). Although seems to facilitate a direct comparison among the healthy/prosthetic leg and the two prosthetic feet, it makes it quite difficult to have an overview of all the results. I would suggest reporting all the results in a table where all the parameters are listed. All the figures can be inserted as Supplementary material. Moreover, if I am not mistaken, in these figures mean values over the 50 gait cycles are reported (while this is clear for the GRF, this is not specified for the other parameters); reporting also the SD might be useful.

Section 2.5.3 (Result section). Several parts reported in this section sound like either a description of the methods or discussing the results – I would suggest rearranging this paragraph accordingly.

Minor comments

Figure 1. The different components of the prosthetic foot are shown. Since in the text the description is organised in foot subgroups, it would be helpful if the different parts of the foot prosthesis are illustrated with different colours.

2.5 Results. I would change the numbering to this section to 3 (i.e., 3. Results); if this change is accepted, all the other section numbering should be changed accordingly.

Five prototypes of MyFlex-δ were produced optimizing 5 stiffness values for as many weight users’ categories: 60 kg, 70 kg, 80 kg, 90 kg and 100 kg. When the prosthetic foot was directly tested on three K4 participants (year of amputation: 10+ years), different stiffness values were tested on each participant. It would be informative to report which stiffness has then been selected for the three analysed participants.

Due to the analysis data from only three participants, the authors did not perform statistical analysis. For this reason, the authors should be careful while reporting the results (e.g., page 10, line 313: “any significant difference”)

Page 15, line 366. Please change from “walking on a horizontal plane” to “level walking”

Author Response

Dear Sir/Ma'am,

please attached you can find our responses.

Thank you very much for reviewing our manuscript.

Best regards.

Reviewer 3 Report

Comments on 1624352 from Applied Science

This study tried to construct an energy stored and releasing ankle/foot system to help gait. The effort is respected but there are many problems on different aspects of study.

Major concerns:

  • Design: The spherical joint may not a good idea for the ankle, as it is different from human anatomy of the ankle complex which has been verified by long time of walking experience in human history.
  • Which variables or measurements did show Energy-Storing-and-Releasing? How much percentages of energy were stored and released?
  • Abstract: ‘there were no significant improvements on the symmetry of the gait…’. If not, why did the authors do this project? Or did this project fail to reach the goal?
  • Introduction: What main points have been changed between this and previous types/generations? Clarify please.
  • Methods: The number of samples is small and thus no statistical conclusion can be made now. This is a serious shortcoming.
  • Figure 6: no significant changes were observed.
  • Figure 15: did you mean ‘step width’ rather than ‘stance’?

Author Response

Dear Sir/Ma'am.

Please attached you can find our responses to your feedbacks.

Thank you very much for reviewing our manuscript.

Best regards.

Round 2

Reviewer 3 Report

authors seem not to solve the questions asked and have not improved the manuscript further

Author Response

Dear Sir/Ma'am,

First of all, thank you for reviewing our manuscript and we apologize for not answering your questions properly. We hope that we could give the proper response in the new response file we attached.

In order to answer your questions, we authors have made further changes to the manuscript. Since we cannot reload the manuscript in the application, we send the file to the assistant editor who is taking care of our work, which we think will forward it to you.

We hope we have improved our work thanks to your feedback and have answered the questions.

Best regards,

the authors.
